# Cocoa Polyphenols Prevent Age-Related Hearing Loss and Frailty in an In Vivo Model

**DOI:** 10.3390/antiox12111994

**Published:** 2023-11-12

**Authors:** Rosalía Fátima Heredia, Juan I. Riestra-Ayora, Joaquín Yanes-Díaz, Israel John Thuissard Vasallo, Cristina Andreu-Vázquez, Ricardo Sanz-Fernández, Carolina Sánchez-Rodríguez

**Affiliations:** 1Department Clinical Analysis, Hospital Universitario de Getafe, Carretera de Toledo, km 12.500, 28905 Getafe, Madrid, Spain; rosaliafatima.heredia@salud.madrid.org; 2Otolaryngology Department, Hospital Universitario de Getafe, Carretera de Toledo, km 12.500, 28905 Getafe, Madrid, Spain; juanignacio.riestra@salud.madrid.org (J.I.R.-A.); joaquin.yanes@gstt.nhs.uk (J.Y.-D.); rsanzf@salud.madrid.org (R.S.-F.); 3Department of Medicine, Faculty of Biomedical and Health Sciences, Universidad Europea de Madrid, 28670 Villaviciosa de Odón, Madrid, Spain; israeljohn.thuissard@universidadeuropea.es (I.J.T.V.); cristina.andreu@universidadeuropea.es (C.A.-V.)

**Keywords:** age related hearing loss, frailty, cocoa polyphenol, antioxidants, total antioxidant capacity, total urinary polyphenol, experimental animals

## Abstract

Age-related hearing loss (ARHL) impairs the quality of life in elderly persons. ARHL is associated with comorbidities, such as depression, falls, or frailty. Frailty syndrome is related to poor health outcomes in old age. ARHL is a potentially modifiable risk factor for frailty. Oxidative stress has been proposed as a key factor underlying the onset and/or development of ARHL and frailty. Cocoa has high levels of polyphenols and provides many health benefits due to its antioxidant properties. Male and female C57Bl/6J mice were randomly assigned to two study groups: animals receiving a cocoa-supplemented diet and the other receiving a standard diet. Then, at the ages of 6, 14, and 22 months, hearing and frailty were measured in all mice. Auditory steady-state responses (ASSR) threshold shifts were measured to different frequencies. The frailty score was based on the “Valencia Score” adapted to the experimental animals. The total antioxidant capacity and total polyphenols in urine samples were also measured. Significant improvements in hearing ability are observed in the cocoa groups at 6, 14, and 22 months compared to the no cocoa group. The cocoa diet significantly retards the development of frailty in mice. Cocoa increases the concentration of polyphenols excreted in the urine, which increases the total antioxidant capacity. In conclusion, cocoa, due to its antioxidant properties, leads to significant protection against ARHL and frailty.

## 1. Introduction

Age-related hearing loss (ARHL), which is also named presbycusis, is defined as the bilateral and progressive deterioration of auditory function [1]. ARHL is the most common sensory disorder affecting the elderly population. The World Health Organization (WHO) estimates that approximately 2.45 billion people worldwide will suffer from ARHL in 2050 [2]. Thus, ARHL presents a major public health concern.

The pathogenesis of ARHL is still not well understood. It is a multifactorial disease to which many risk factors contribute, including extrinsically (exposures to environmental ototoxic agents, noise, immune system, vascular injury, trauma, hormones, metabolic changes, and diet), and intrinsically (genetic and physiological aging processes) [3,4,5,6]. Ultimately responsible for ARHL is the death of sensory cells and spiral ganglion (SG) neurons in the cochlea (hearing organ). So far, no successful treatment has been found for this age-related pathology.

ARHL is reported to increase the risk of falls, depression, cognitive decline, dementia, and frailty in older adults. ARHL is a significant marker for frailty in older age [7], another age-related multidimensional clinical syndrome characterized by a reduced multisystem physiological reserve, a nonspecific state of vulnerability, and decreased resistance to stressors that expose older persons to an increased risk of falls, disability, hospitalization, institutionalization, and death [8]. Recent research data have demonstrated that frailty is not a static syndrome and many of its symptoms and signs can be reversed partially with suitable interventions, such as a review of medications, diet, and physical activity [8,9,10,11]. These promising findings have led to new efforts to uncover other potentially alterable risk factors related to frailty, and ARHL is one of them. Several potential pathways link ARHL with frailty [12], among which oxidative stress is highlighted [13].

The free radical theory postulates that aging is the result of accumulated or increased oxidative damage caused by free radicals [4,6,14]. Free radicals are essentially reactive oxygen and nitrogen species (ROS/RNS), an inevitable product of cellular respiration. When the endogenous antioxidant system is overcome either by the production of excess ROS/RNS, accumulation of toxic free radicals occurs, leading to oxidative-stress-induced damage to proteins and lipids in the cytosol and cell membranes, as well as to the mitochondrial and nuclear genome. Age-related oxidative stress is an underlying risk factor for many age-related disorders, such as cancer, diabetes, Parkinson’s disease, cardiovascular disease, Alzheimer’s disease, and ARHL [15,16,17]. Recently, frailty and related oxidative stress have drawn significant attention [7,18,19].

The antioxidant defense network plays a crucial role in the protection of the organism against ROS/RNS. Based on their activity, antioxidants can be classified as enzymatic and nonenzymatic, and these, in turn, into exogenous (such as phytochemicals) and endogenous (such as vitamins) [16,17,18]. Under normal conditions, ROS/RNS are metabolized or scavenged by enzymatic antioxidant mechanisms (e.g., catalase [CAT], glutathione peroxidase [GPx], and superoxide dismutase [SOD]). However, the aging process alters this homeostatic condition in the hearing organ. Regarding non-enzymatic exogenous antioxidants, a wide variety has been used, such as melatonin, resveratrol, L-ascorbic acid, L-carnitine, N-acetylcysteine, quercetin, etc. In our study, we will use cocoa for its antioxidant properties [16,17,18].

Supporting this, antioxidant-based treatments should reduce hearing loss induced by oxidative stress. Accordingly, several antioxidants have proven to be effective, mainly at the level of proof of principle, for the treatment of ARHL [20,21]. This confirms the pivotal role that oxidative stress plays in the development and/or progression of ARHL.

Epidemiological research has shown that high plasma concentrations of antioxidants, such as vitamins E and D and carotenoids, are correlated with a lower prevalence of frailty and associated conditions [22,23]. Indeed, high levels serum of carotenoids were related to a reduced risk of sarcopenia and walking disabilities, and low carotenoid concentrations have been considered a risk factor for accelerated muscle strength decline [24,25,26].

Cocoa beans from the Theobroma cacao (cocoa tree) plant are a higher source of polyphenols (specifically, flavanols). Cocoa-obtained products are commonly consumed in many countries in the world [27,28]. Many studies have shown that cocoa has beneficial properties against oxidative-stress-related diseases by increasing the activities of antioxidant enzymes and scavenging free radicals [27,28]. Thus, cocoa can be considered a natural product that provides dietary antioxidants with therapeutic properties. However, to our knowledge, the preventive effects of cocoa polyphenols on ARHL and frailty has not yet been investigated.

In this research, we show that the oxidative stress linked to aging could be, at least in part, primarily involved in the interplay between ARHL and frailty. Furthermore, to date, there is no evidence of cocoa’s usefulness for the preventive treatment of ARHL and frailty. Our study aimed to determine whether the intake of flavanol-rich cocoa halted or attenuated ARHL and frailty syndrome in an aging murine model.

## 2. Materials and Methods

### 2.1. Experimental Animals

The animal model chosen for this study was C57BL/6 mice since it is an accepted model of ARHL [29]. Adult male and female C57BL/6J mice (*n* = 100), 3 months old, supplied by Charles River Laboratories (Wilmington, MA, USA) were maintained in conditions of 22–25 °C temperature, 50% relative humidity, and 12 h day/night cycles. Food and water were available ad libitum. Mice were checked every day. The Ethical Committee for Animal Experimentation of Comunidad de Madrid (PROEX 60.1/20) approved all procedures.

The animals were acclimatized for 7 days before the experiment and then randomly assigned to one of two groups: the control group (*n* = 47) or the cocoa group (*n* = 47), the latter’s diet supplemented with cocoa at 8.2 mg/kg body weight (b.w.). For this study, cocoa powder with a high content of polyphenols (Chococru, London, UK) was used. Cocoa doses were equivalent to the daily intake of 5 g cocoa for a 70 kg person according to an allometric scale [30]. The cocoa was mixed with the usual feed at 8.2 g/kg supplied by Safe Diets (Augy, France). The diet provided an approximate daily intake of 25 mg of cocoa (considering that they consumed ca. 3 g of pellet per day).

When the mice reached 6 months of age, 24 mice (male and female, 12 control group, 12 cocoa group) were tested for hearing and frailty, creating the 6-month study group. After completing the tests, the 6-month-old group was sacrificed. This same procedure was performed for the 14- (24 male and female mice, 12 control group, 12 cocoa group) and 22-month-old (46 male and female mice, 23 control group, 23 cocoa group) groups (Figure 1). Six mice died during the study period. The study began with a total of 100 mice and ended with 94.

### 2.2. Auditory Steady-State Response Measurements

The auditory steady-state response (ASSR) was performed on all the mice in the study prior to sacrifice. The investigator who performed the measurements was blinded to the experimental group. The tests ASSR conducted on a single animal were performed during the same session by the same investigator. Briefly, a mouse was anesthetized with an intraperitoneal injection of a mixture of ketamine (0.75 mL/100 g) and medetomidine (0.5 mL/100 g). Body temperature was maintained at 37.5 °C with a warming blanket (Aposán S.A., Madrid, Spain). An ER3 earphone was inserted directly into the external auditory canal. The positive subcutaneous electrode was placed over the vertex and the reference electrodes were placed in the pinna of each ear. Ground electrodes were placed over the biceps femoris muscle (Figure 2).

The ASSRs were recorded using an evoked potential averaging system, the Intelligent Hearing System Smart-EP (Miami, FL, USA), in an electrically shielded, double-walled, sound-treated booth in response to tone bursts at 4, 8, 12, 16, and 32 kHz with a 10 ms plateau and a 1 ms rise/fall time. Intensities were expressed in decibels of the sound pressure level (dB SPL) peak equivalent. Intensity series were recorded, and an ASSR threshold was defined as the lowest intensity capable of eliciting a replicable, visually detectable response. A total of 250 sweeps were averaged at each stimulus level. For the hearing analysis, the mean response to tone bursts in the left ear and the right ear of each animal was calculated.

### 2.3. Body Weight

Mouse body weight was recorded monthly by using a PB3002 Delta Range balance (Mettler Scales, Toledo, OH, USA) group.

### 2.4. Motor Coordination Test

To measure motor coordination in mice, we relied on the tightrope test previously described [31]. Briefly, the mice were deposited in the middle of a 60 cm long and 1.5 cm wide rope. The tests were successful if the mouse reached the end of the rope or was maintained on it for 60 s. All the animals had five opportunities to finish the test. We determined the percentage of mice that succeeded in passing the test (Appendix A).

### 2.5. Grip Strength Test

The neuromuscular function of the mice was determined using the Grip Strength Meter (Panlab. Harvard Apparatus, Cornellá de Llobregat, Spain) by detecting the peak amount of force that the mice applied in grasping the pull bar of the meter.

All mice performed the test. The animals held on to the pull-bar with the forelimb for a few seconds, while the researcher gently pulled the tail away from the sensor (Appendix A). The apparatus automatically recorded the maximum force in grams-force. Immediately, for each mouse, four additional trials were performed [32].

### 2.6. Incremental Treadmill Test

The animals were submitted to a graded intensity treadmill test (Rodent Treadmill NG, Ugo Basile, Gemonio, Italy) to determine their endurance (running time) and running speed along the study. The exercise protocol was established according to the previous protocol of Gómez-Cabrera et al. [33]. After a warm-up period, the treadmill band velocity was increased until the animals were unable to run further.

The initial bout of 6 min at 6 m/min was followed by consecutive 2 m/min increments every 2 min. Exhaustion was defined as the third time a mouse could no longer keep pace with the speed of the treadmill and remained on the shock grid for 2 s rather than running. Exercise motivation was provided for all rodents by means of an electronic shock grid at the treadmill rear. However, the electric shock was used sparingly during the test. We recorded the running time (endurance) and the maximal running speed achieved by the mice (Appendix A).

### 2.7. Frailty Score

The frailty score was established according to the previous protocol of Gomez-Cabrera et al., 2017 [33]. This score, called the “Valencia Score”, is based on the previous one for frailty developed for humans by Linda Fried et al., 2001 [8], but it has been adapted to experimental animals. The frailty score consists of the measurement of five components: body weight (weight loss), grip strength (weakness), treadmill test (low endurance and speed), and motor coordination (poor activity level).

The reference value of body weight for all the animals studied was obtained as follows: the mice for each age group were weighed. The weights were averaged for each group, and this was 100% of the weight of a mouse. When an animal lost more than 5% of the weight at the corresponding age, it was considered positive for the frailty criteria of body weight.

For the grip strength component, we fixed the 20th percentile as a cut-off point according to the previous study of Gomez-Cabrera et al. (2017) [33]. Mice that were graded below the 20th percentile complied with the frailty criteria of weakness. For example, at 6 months of age, the 20th percentile was 78.3 g. Those mice that classed below 78.3 g indicated weaker strength and were regarded as positive for this frailty criterion at that age. We made these calculations for 14- and 22-month-old mice.

As previously described, we established the 20th percentile as each age group’s cut-off point for endurance and maximum running speed. Those animals that reported less than the calculated threshold for each test were positive with this frailty criterion.

The frailty score for each age group of animals was calculated as follows: total number of tests failed by the animals at each age group (A), divided by the total number of tests performed by these animals (B), expressed in percentage [(A/B) × 100].

### 2.8. Urine Collection Method: Hydrophobic Sand

We used hydrophobic sand, LabSand (Coastline Global Inc., Palo Alto, CA, USA), for the collection of small urine samples in the mice. It is a single-use product made from sterilized sand and treated with a hydrophobic coating. The urine sample remains on the top of the sand in droplet form and is easily collected with pipettes.

Mice were in a cage with the LabSand lining the bottom of the cage in place of regular bedding; urine pooled on top of the sand and was then pipetted into a sterile tube and stored at −80 °C until use. Urine was collected for each age group at 6, 14, and 22 months of age. For each collection time, two 6 h collection sessions were performed separated by a rest period of at least 48 h. Urine samples from each animal were pooled for subsequent analysis.

During urine collection sessions, water and food were not provided to any mice, but each one was provided with a water replacement gel in a plastic bowl (HydroGel^®^, Clear H_2_O, Westbrook, ME, USA).

### 2.9. Determination of Total Polyphenols in Urine Samples

The determination of total polyphenols in baseline urine samples was performed BQC Phenolic Quantification Assay Kit (Bioquochem, Oviedo, Spain) is based on the Folin–Ciocalteau spectrophotometric method [34] after solid-phase extraction using Oasis MAX 96-well plates (Waters, Milford, MA, USA) according to the manufacturer’s instructions. The kit uses gallic acid as standard (0 to 300 µg/mL) and employs the FLUOstar Omega (BMG Labtech, Ortenberg, Germany) plate reader to measure it (OD 700 nm).

### 2.10. Measurement of Total Antioxidant Capacity

The total antioxidant capacity (TAC) was measured by the e-BQC portable device (Bioquochem; Oviedo, Spain) in the plasma of mice. The system is based on the measurement of redox potential (charge/period or micro-Coulomb, μC). The samples (30 µL) were dispensed onto a disposable strip. System readings were given for rapid (Q_A_), slow (Q_B_), and total (Q_T_: Q_A_ + Q_B_) antioxidant responses.

### 2.11. Statistical Analysis

The descriptive analysis was conducted for control and cocoa groups and separately for animals aged 6, 14, and 22 months within each group. The quantitative variables (weight, results of grip strength test, treadmill test (endurance and speed), motor coordination test, and hearing threshold values for 4 kHz, 8 kHz, 12 kHz, and 16 kHz) are described using the median and interquartile range (Q1–Q3) after confirming that they did not follow a normal distribution (Shapiro–Wilk test). The qualitative variables (frailty score and its five components) are described using absolute frequencies (*n*) and relative frequencies (%).

The comparison of quantitative variables at 6, 14, and 22 months of age between the animals in the control and cocoa groups was carried out using the Mann–Whitney U test. To compare these variables between animals with different ages in the same study group, Kruskal–Wallis test and post hoc Sidak test were used for pairwise comparisons when significant differences were detected. The chi-square test or Fisher’s exact test were used to compare between groups the proportion of animals failing each of the five components of the frailty score at each age. Equivalence tests of proportions were used to determine whether there were differences or not in the frailty score between control and cocoa animals at different ages. The analyses were conducted using Stata BE version 17 (StataCorp LLC, College Station, TX, USA). Statistical significance was determined at a *p*-value threshold of less than 5%.

Results for total polyphenols in urine and total antioxidant capacity measurements are expressed as mean ± SD of individual samples of the ‘‘n/N’’ numbers given. The statistical analysis was performed using the SPSS 19.0 software (IBM, Armonk, NY, USA). Differences in the mean or variance were evaluated using the factorial analysis of variance (ANOVA), followed by Fisher’s protected least significance test with the significance level chosen at *p* < 0.05.

## 3. Results

### 3.1. Cocoa Diet Prevents Age-Related Hearing Loss

At 6 months of age, ASSR threshold shifts were analyzed. At this time point, the cocoa group had received three months of antioxidant therapy. Statistically significant differences in the threshold between the control versus treatment groups were found for all frequencies studied (Table 1 and Figure 3).

The 14-month age group ASSR data are presented in Table 1 and Figure 1. At this time point, the cocoa group had received eleven months of antioxidant therapy. The average threshold for the control group was found to be 70 dB, and for the cocoa group 50 dB, at 32 kHZ (*p* < 0.001). Similarly, there was a significant difference between thresholds for the control versus the cocoa treatment groups at 4, 8, 12, and 16 kHz (*p* < 0.001).

Finally, the ASSR thresholds of the 22-month-old group were measured (Table 1 and Figure 3). At this time point, the cocoa group had received 19 months of antioxidant therapy. A statistically significant difference was noted in ABR thresholds of the control versus cocoa groups at 4, 12, 8, 16, and 32 kHz. At 12 kHz frequency, e.g., the threshold for the cocoa group was found to be 45 dB, and for the control group it was 65 dB (*p* < 0.001).

### 3.2. Cocoa Prevents Loss of Body Weight

Animals’ body weights were recorded at 6, 14, and 22 months. In the control group, a lower weight was observed in the 22-month-old animals compared to the 6-month-old animals (31.5 (20.25, 33.25) g vs. 27.5 (25.5, 30) g, *p* = 0.001); this weight loss was not evidenced in the animals of the cocoa group (*p* = 0.389) (Table 2 and Figure 4). As the animals grew older, the mice of the cocoa group lost less weight than the control ones. At 14 and 22 months, the mice in the cocoa group weighed more than those in the control group (Table 2 and Figure 4).

### 3.3. The Cocoa Diet Improves the Grip Strength in Aged Mice

We measured the grip strength of the animals at ages of 6, 14, and 22 months. Figure 5 shows the increase in weakness in the mouse grip strength test associated with age; these differences were statistically significant in the control group (*p* = 0.003, 6 months vs. 14 months and *p* < 0.001, 6 months vs. 22 months). However, the animals of different ages belonging to the cocoa group showed a tendency toward greater grip strength than the controls. Grip strength was lower at 14 months in the control group than in cocoa group (*p* = 0.013) (Table 3 and Figure 5).

### 3.4. The Cocoa-Rich Diet Ameliorates Motor Coordination in Aged Mice

The tightrope test measured neuromuscular coordination (an index of motor coordination and physical strength). The test quantified the percentage of animals that successfully passed the tightrope test. We determined the motor coordination at the ages of 6, 14, and 22 months. Figure 4 shows that age-advanced control groups (14- and 22-month-old), with fewer passing members, had worse results than the 6-month control group in the test. On the contrary, the percentage of success in the tightrope test was significantly higher for the cocoa groups at 14 and 22 months of age with respect to the control groups. In the 6-month-old group, no significant differences were found between the control and cocoa group (Figure 6).

### 3.5. The Cocoa Diet Increases Endurance and Running Speed in Older Mice

We determined the endurance of animals by measuring the running time when performing an incremental intensity test on a treadmill in all age groups studied. Table 4 and Figure 7A shows that mice in the 6-month-old control group performed significantly better in the running time test than 14- and 22-month-old mice in the control groups. However, we found that the cocoa-rich diet significantly improved the endurance of mice aged 14 (*p* = 0.003) and 22 months (*p* = 0.004) compared to their respective control groups (Table 4 and Figure 7A).

In addition, we measured the maximum running speed achieved when performing an incremental intensity test as a marker of sluggishness (Table 4 and Figure 7B). Table 4 and Figure 7B shows that the control groups of 14- and 22-month-old mice performed worse on the incremental run test than the control group of 6-month-old mice, while in mice with a diet rich in cocoa, for all ages studied, maximum running speed was significantly improved (Figure 7B and Table 4).

### 3.6. Cocoa Decreased Frailty in Elderly Mice

Finally, Table 5 shows the failures for each test/parameter (weight, grip force, motor coordination, speed, and endurance) and the percentage frailty score calculated with the data collected for each parameter by applying the formula previously described. The results show that, in the mice of the control groups, as they age, the percentage of frailty increases, being significantly higher in the 22-month-old group than in the 6-month-old group (*p* < 0.001) (Figure 8). On the contrary, as shown in Table 5 and Figure 6, the 22-month-old mice with a diet rich in cocoa present a lower frailty score than the control group mice (*p* = 0.002).

### 3.7. The Cocoa Diet Increased the Total Antioxidant Capacity in Older Mice

Figure 9 showed the results of total antioxidant capacity (TAC) Q_T_ as Q_A_ (fast antioxidants) and Q_B_ (slow antioxidants). Q_A_, Q_B_, and Q_T_ were higher in the 6-month group (control and cocoa diet) than the 14- and 22-month control group; therefore, the elderly mice exhibited lower TAC than the 6-month control and cocoa groups. The cocoa intake induced statistically significant recovery of antioxidant capacity in 14- and 22-month groups (*p* < 0.05) (Q_T_: Q_A_ + Q_B_).

### 3.8. Cocoa Intake Increases Urinary Total Polyphenols

We also checked urinary total polyphenols (UTP) (as a biomarker of total dietary polyphenols (TDP) intake). UTP levels were significantly increased after the intake of cocoa in all age groups compared with standard diet groups and as expected, levels of UTP in groups that received a cocoa diet were significantly higher than in their respective controls (Figure 10). UTP levels did not change in control groups (Figure 10).

## 4. Discussion

ARHL negatively affects the quality of life of older people, posing an important health problem for public health systems worldwide. In aging, ARHL is also a relevant marker for frailty, another age-related multidimensional clinical state with a diminished multisystem physiological reserve, nonspecific condition of vulnerability, and reduced resistance to different stressors (such as injuries, sensorial impairments, diseases, and psychosocial stress) that associates older people with an increased probability of falls, institutionalization, hospitalization, and, finally, death [7,8].

Oxidative stress, among other mechanisms, has been proposed as playing a principal role in the development of aging. Oxidative damage is a crucial component in pathological pathways that are thought to drive several age-related pathologies [35]. To date, various studies suggest that oxidative stress is involved in the pathophysiology of ARHL and frailty [13,17]. Therefore, antioxidant therapy can slow ARHL and frailty syndrome.

Cocoa has been known for its health benefits for centuries. The discovery of biologically active polyphenol compounds in cocoa has stimulated research on its effects on oxidative stress, aging, atherosclerosis, blood pressure regulation, and other diseases. Today, cocoa is used for its great antioxidant potential. Cocoa contains high concentrations of epicatechin, flavonoids, procyanidins, and catechin. Cocoa has greater levels of flavonoids than wine and tea [36]. The present study showed the potential health benefits of cocoa polyphenols (due to antioxidant properties), which lead to significant protection against ARHL and frailty.

Prevention therapies for ARHL are a relatively new area of research. In the last years, numerous studies have been conducted to prevent or slow the onset or development of ARHL, which have mainly included treatment with exogenous antioxidants. In our study, we demonstrate that a cocoa-enriched diet has protective effects on ARHL. These findings indicate significantly improved ASSR auditory thresholds in mice receiving treatment with cocoa at the frequencies studied (from 4 to 32 kHz), supporting a possible relationship between the hearing data and the supposed protection of the cochlear sensory tissue. With this study, for the first time, we demonstrate the otoprotective effect of cocoa against ARHL.

This research appears to agree with previous studies. Several studies that used animals have demonstrated that exogenous antioxidants can prevent ARHL. Seidman et al. (2002) showed the protective effects of lecithin on ARHL [37]. Heman-Ackah et al. [38] found that a combination of antioxidant (ascorbic acid, L-cysteine-glutathione mixed disulfide, ribose-cysteine, vitamin B12, ribose-cysteine, N-ω-nitro-L-arginine methyl ester, and folate) successfully decreased the threshold shifts of the ABR in C57BL/6 mice. Previously, our group offered results supporting the protective benefits of antioxidant EGb761 in hearing in Sprague–Dawley old rats (SD) [39]. Also, in SD rats, we demonstrated that a mixture of polyphenols (resveratrol, tannic acid, quercetin, morin, rutin, and gallic acid) significantly improved hearing levels in old rats [40]. Other agents, such as N-acetyl-L-cysteine (NAC) [41], α -lipoic acid [42], and CoQ10 [43], have also been studied individually for preventing ARHL.

However, whether the progression of ARHL is sensitive to antioxidant therapy is controversial. Some studies have failed to detect a protective effect of antioxidants in ARHL. Sha et al. [44] showed that diet supplementation with L-carnitine, α-lipoic acid, and vitamins A, C, and E does not ameliorate hearing loss in CBA/J old mice (24 months of age). Similarly, Davis et al. [45] did not confirm the protective effect of NAC (strong antioxidant) in the aged C57BL/6J mice. Also, acetyl-carnitine (ALCAR) failed to alter ARHL in Fischer 344/NHsd rats [46].

Oxidative stress, either due to lack of elimination of or overproduction of ROS, is pivotal not only for the development of ARHL to start but also for triggering frailty syndrome. Therefore, the reduction of oxidative stress for antioxidant therapy could be an important objective in the prevention of frailty syndrome. Our study was the first to show that cocoa-rich diet interventions can delay frailty in mice. It not only provides preclinical evidence for exploring the effect of these interventions in humans but also offers the value of the mouse clinical frailty model as a preclinical tool for assessing possible interventions for frailty. Previous studies are in line with our results, e.g., Kane et al. 2016 [47] in a mouse model of frailty showed that resveratrol treatment delayed frailty in mice. Frailty syndrome was measured in old (18 months) male and female C57BL/6 mice. Resveratrol (100 mg/kg) significantly lowered the frailty score in mice fed the resveratrol diet compared to a control.

Many studies revealed that flavonoids from cocoa [48] exert beneficial effects on the brain. Emerging evidence has suggested that flavonoids protect against degeneration in dementia and Alzheimer’s disease. The effects of cocoa on animal cognitive function were previously described. One study investigated the long-term effect of a cocoa polyphenolic extract, Acticoa powder. The authors concluded that a 12-month oral administration of this powder in aged rats showed improved cognitive performance in terms of short- and long-term learning and spatial memory [49]. Moreira et al. 2016 [50], in a longitudinal prospective study with a cohort of 531 participants aged 65 and over with normal MiniMental State Examination (MMSE; median 28), showed that long-term chocolate consumption is associated with a lower risk of cognitive decline in humans.

The protective effect of polyphenols against fragility could be due to their antioxidant and anti-inflammatory activity [51,52,53]. Greater adherence to the Mediterranean diet (rich in fruits and vegetables) was related to lower levels of inflammatory markers, for example, CRP and IL-6, especially those related to endothelial function (VCAM-1 and ICAM-1) [53,54]. Furthermore, it has been hypothesized that diets rich in antioxidants (some vitamins and polyphenols) could prevent sarcopenia, a major element of frailty [51]. Diets containing antioxidants produced elevated glutathione activities and less oxidative stress in the mitochondria and muscles of mice [55] and protected skeletal muscle from oxidative damage [56].

Also, our study finding that high concentrations of UTP were associated with a lower prevalence of ARHL and frailty in older mice. The effect of polyphenols on health depends on their bioavailability and the amount consumed; this varies greatly depending on individuals and the type of polyphenol [57,58]. For this reason, a biomarker of total dietary polyphenols (TDPs) is needed to precisely assess the relationship between chronic diseases and total polyphenols. Urinary total polyphenol (UTP) measured by the Folin–Ciocalteu urine assay is considered a valid nutritional biomarker for TDP and a representative biomarker of dietary vegetable and fruit intake [59,60]. Polyphenol markers have some advantages over dietary data obtained by self-reported questionnaires [53]. The principal advantage of dietary markers is that they supply an objective measure of exposure that is independent of many of the errors and biases associated with self-report procedures [54]. In previous study, Zamora-Ros et al. (2013) [61] demonstrated in a population of older adults that high concentrations of a urinary biomarker of polyphenol intake were associated with a decrease in mortality. In another study, high levels of UTP were related to a lower prevalence of prefrailty and frailty in an older population, concluding that a diet rich in polyphenols may protect against frailty in older people [62].

Finally, we demonstrated that the total antioxidant capacity (TAC) in the plasma of mice was increased by a diet rich in cocoa. These results align with UTP values in urine, which are higher in mice that ingest cocoa. These data show that a diet rich in polyphenols improves the antioxidant capacity and, therefore, the antioxidant defense of the body, which implies, in this case, an improvement in hearing and a lower frailty score.

Dietary total antioxidant capacity (TAC) has been a useful tool in several studies, e.g., the research by Diao et al. 2003 [63] aimed at studying the effect of noise on total antioxidant capacity (TAC) in serum, nitric oxide (NO) levels in the cochlea, and the protective effect of alpha-lipoic acid against noise-induced hearing loss (NIHL) in guinea pigs. The data obtained indicate that noise causes a decrease in serum TAC and an increase in NO in the cochlea. The TAC level of the noise+alpha-lipoic acid group was significantly higher than that of the noise+saline group. Alpha-lipoid acid has a protective effect against hearing loss through its antioxidant effects. In another study, Tembo et al. (2020) [64] investigated the association between frailty in older men and serum total antioxidant capacity (TAC). These data propose a positive association between frailty and TAC levels, supporting the theory that this marker could be useful in identifying people at risk of frailty.

## 5. Conclusions

In conclusion, the findings of our study suggest that a diet rich in cocoa prevents ARHL and frailty syndrome in an aging murine model. This is the first study to investigate the beneficial effect of a cocoa diet on these aging diseases. However, due to the limitations of in vivo models (preclinical), clinical studies will be necessary to provide a clinical approach for patients.

## Figures and Tables

**Figure 1 antioxidants-12-01994-f001:**
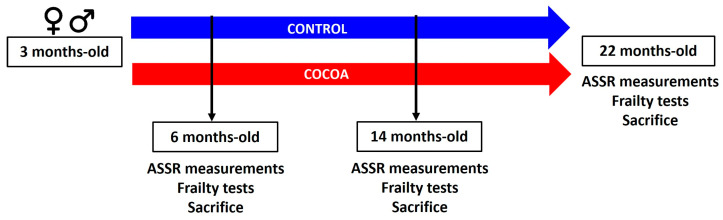
Comprehensive overview of the experimental timeline and procedures.

**Figure 2 antioxidants-12-01994-f002:**
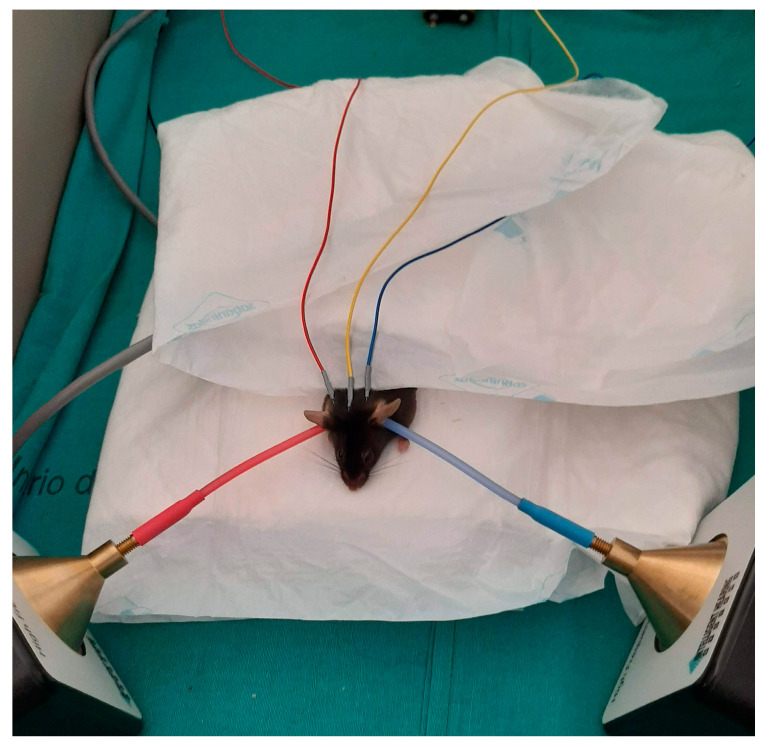
Auditory steady-state response measurements in mice.

**Figure 3 antioxidants-12-01994-f003:**
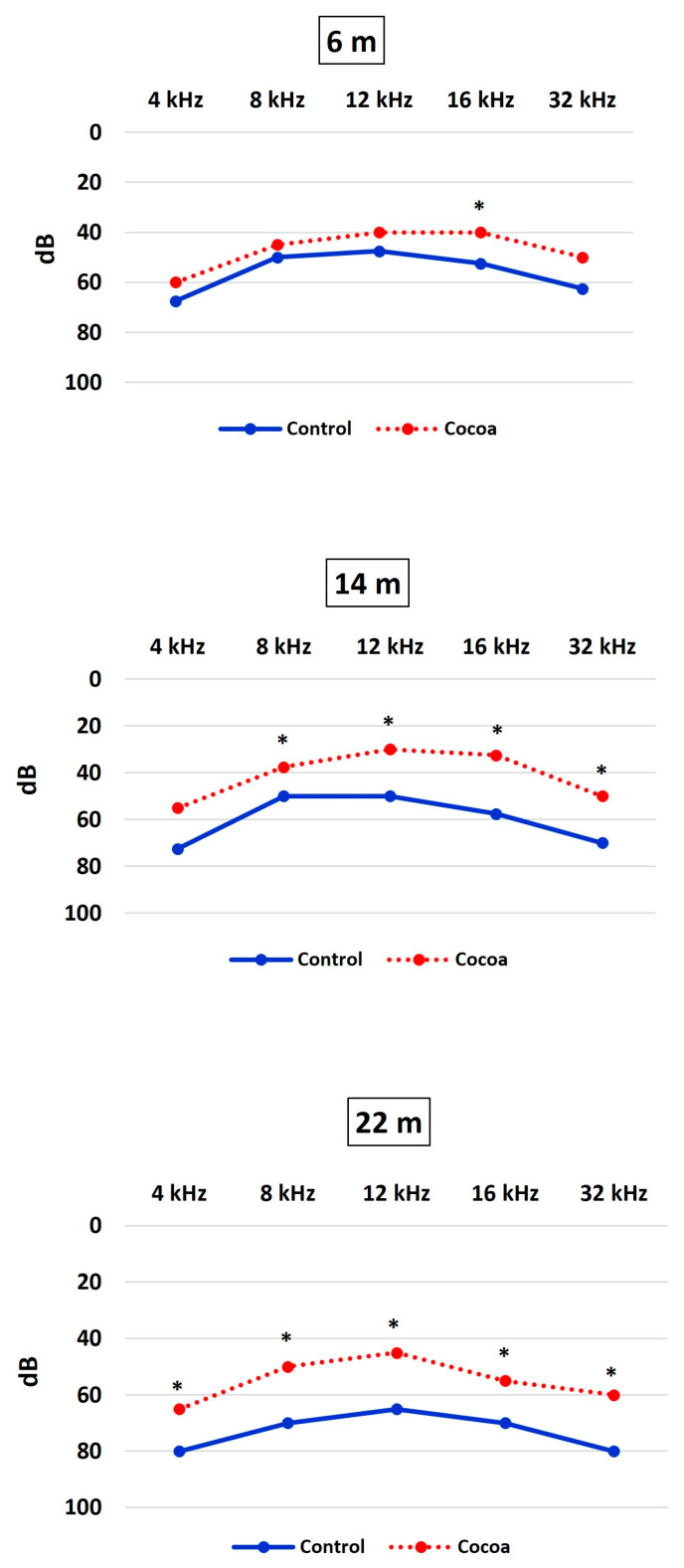
The cocoa diet prevents age-related hearing loss. The ASSR threshold shifts within frequencies (4–32 kHz) were measured in C57Bl/6J mice at different ages (6 months, 14 months, and 22 months) and treatments (control and cocoa diet). Data are expressed as median ± SD of hearing levels expressed in decibels of the sound pressure level (dB), asterisks indicate significant differences (*p* < 0.05) in between-group comparisons at each time (evaluated using Student’s *t*-test).

**Figure 4 antioxidants-12-01994-f004:**
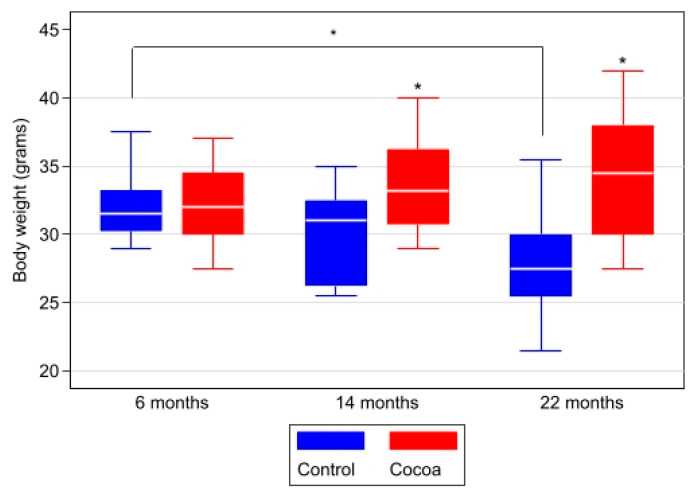
Cocoa prevents age-related loss of body weight. Animals’ body weights were recorded at 6, 14, and 22 months. Data are expressed as median (Q1, Q3) of body weight in grams. Box-plot graph with medians, Q1, Q3, and asterisks indicating significant differences (*p* < 0.05) in between-group comparisons at each time (evaluated using Mann–Whitney–Wilcoxon test) and within-group between moments (evaluated using Kruskal–Wallis test and post hoc Sidak test for pairwise comparisons).

**Figure 5 antioxidants-12-01994-f005:**
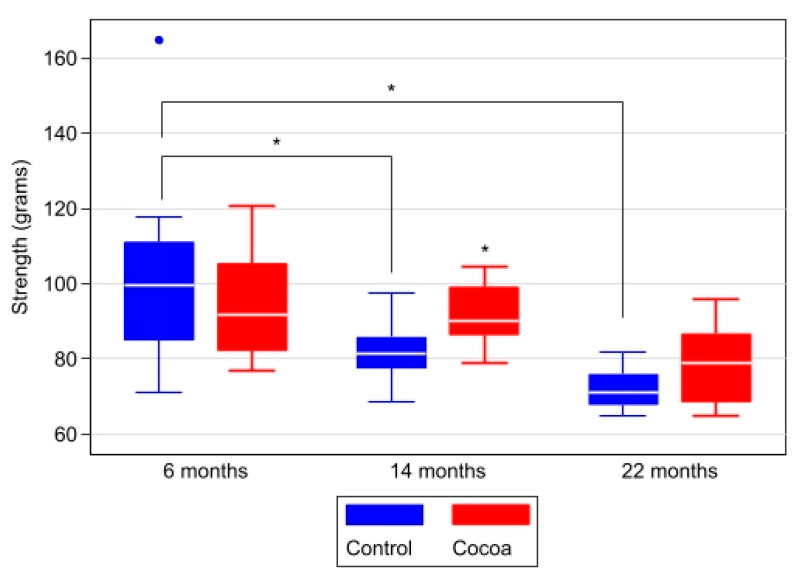
The cocoa diet improves the grip strength in aged mice. Reference mice grip strength values at different ages were obtained. Data are expressed as median (Q1, Q3) of grip strength in grams. Box-plot graph with medians, Q1, Q3, and asterisks indicating significant differences (*p* < 0.05) in between-group comparisons at each time (evaluated using Mann–Whitney–Wilcoxon test) and within-group between moments (evaluated using Kruskal–Wallis test and post hoc Sidak test for pairwise comparisons).

**Figure 6 antioxidants-12-01994-f006:**
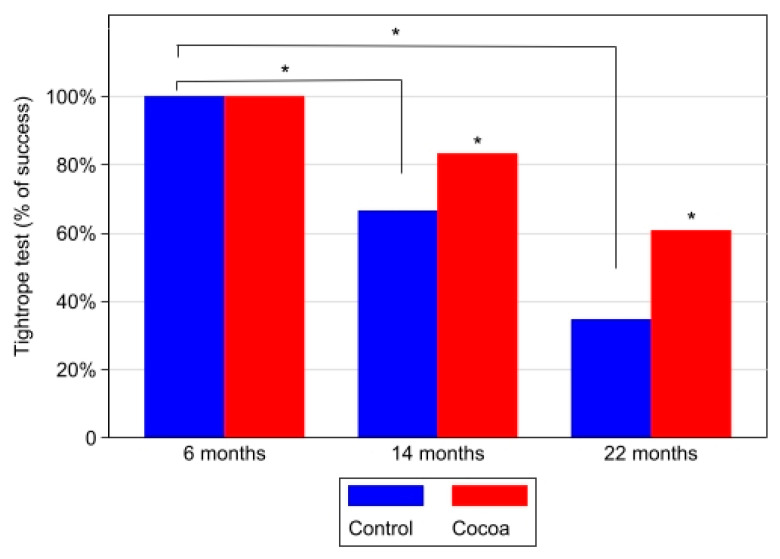
The cocoa-rich diet ameliorates motor coordination in aged mice. It was determined as the percentage of animals that successfully passed the tightrope test at different ages (6, 14, and 22 months). Box-plot graph with medians, Q1, Q3, and asterisks indicating significant differences (*p* < 0.05) in between-group comparisons at each time (evaluated using Mann–Whitney–Wilcoxon test) and within-group between moments (evaluated using Kruskal–Wallis test and post hoc Sidak test for pairwise comparisons).

**Figure 7 antioxidants-12-01994-f007:**
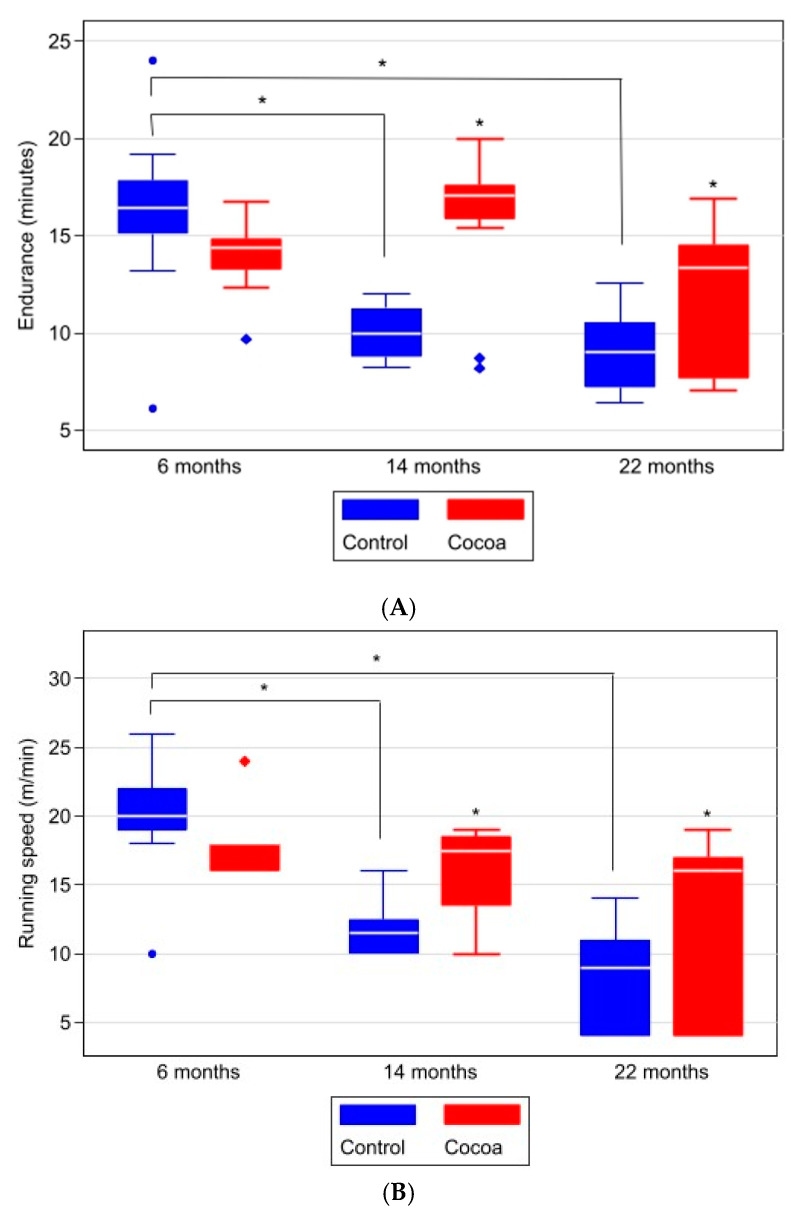
The cocoa diet increases endurance and running speed in older mice. We determined the running time (**A**) and running speed (**B**) values at different ages in mice. Data are expressed median (Q1, Q3) of body weight in grams. Box-plot graph with medians, Q1, Q3, and asterisks indicating significant differences (*p* < 0.05) in between-group comparisons at each time (evaluated using Mann–Whitney–Wilcoxon test) and within-group between moments (evaluated using Kruskal–Wallis test and post hoc Sidak test for pairwise comparisons).

**Figure 8 antioxidants-12-01994-f008:**
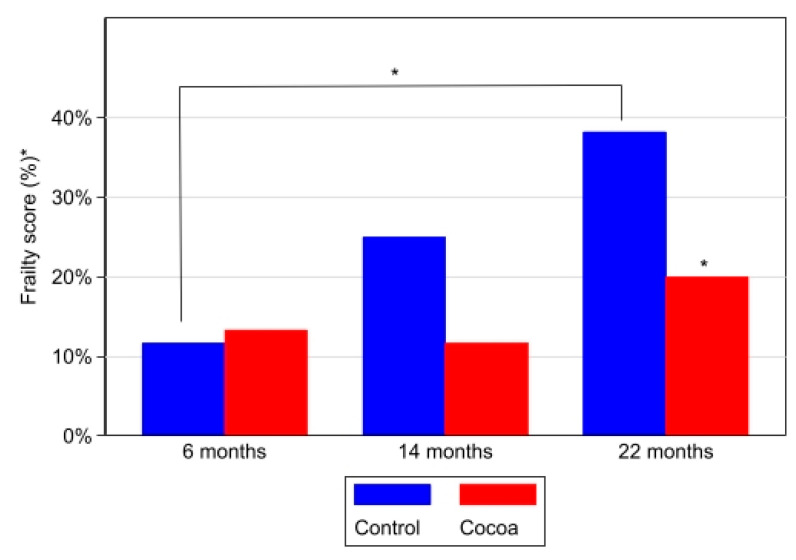
Cocoa decreased frailty in elderly mice. The frailty score for each age group of animals was calculated as follows: the total number of tests failed by the animals at each age group was divided by the total number of tests performed by these animals and expressed as a percentage. Box-plot graph with medians, Q1, Q3, and asterisks indicating significant differences (*p* < 0.05) in between-group comparisons at each time (evaluated using Mann–Whitney–Wilcoxon test) and within-group between moments (evaluated using Kruskal–Wallis test and post hoc Sidak test for pairwise comparisons). * Total number of tests failed by animals at each age group, divided by the total number of tests performed by these animals.

**Figure 9 antioxidants-12-01994-f009:**
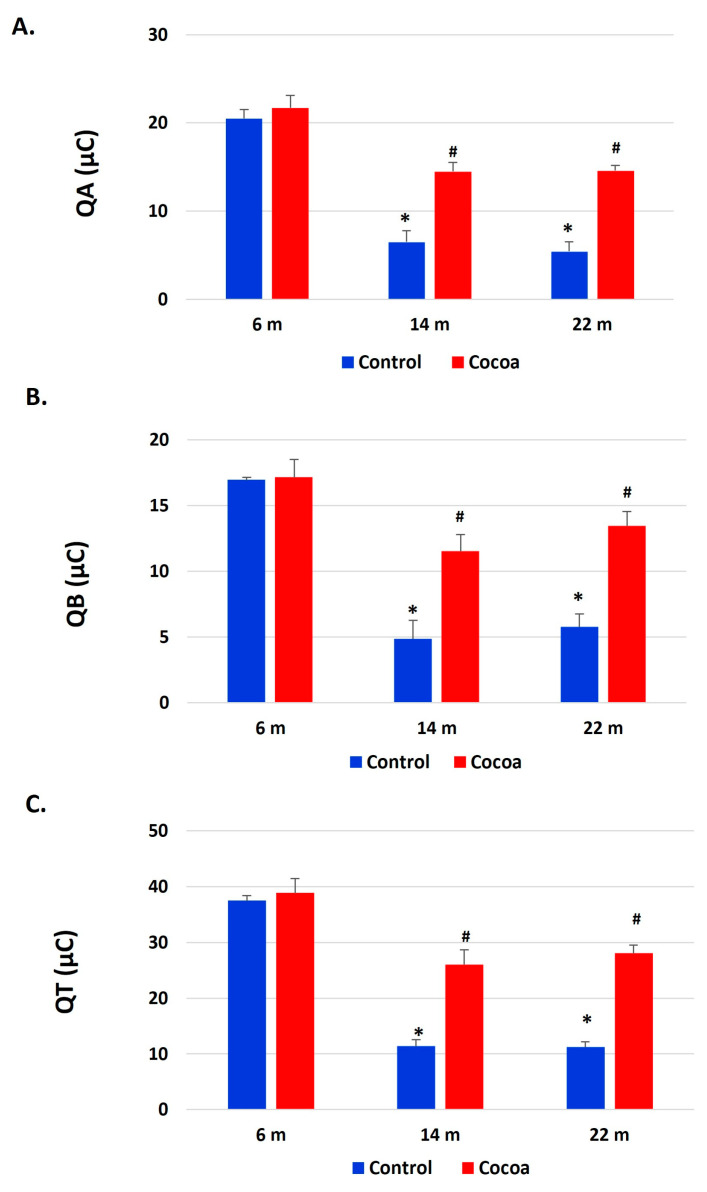
Cocoa treatment modulated total antioxidant response. (**A**) Q_A_ (fast antioxidants), (**B**) Q_B_ (slow antioxidants), and (**C**) Q_T_ (total antioxidant response: Q_A_ + Q_B_) (µC) were measured by e-BQC lab system in plasma of mice at different ages (6 months, 14 months, and 22 months). The values represent the mean ± SD of micro-Coulomb (µC). * *p* < 0.05 vs. 6-month-old control group; ^#^
*p* < 0.05 vs. their respective control (evaluated using Student’s *t*-test).

**Figure 10 antioxidants-12-01994-f010:**
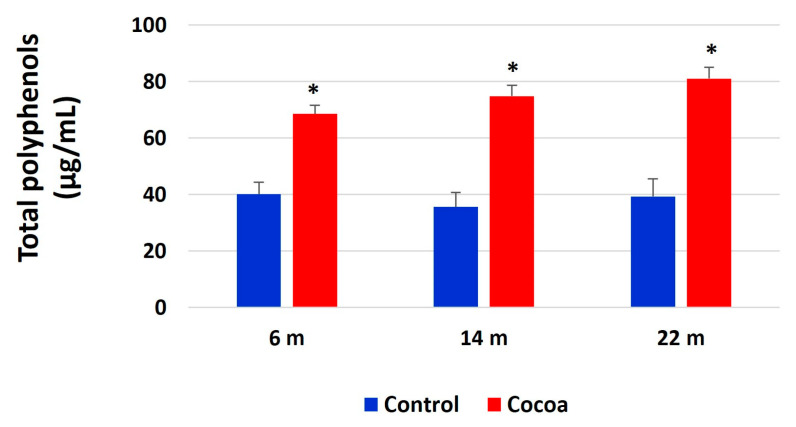
Cocoa intake increases total polyphenols in urine. Total polyphenols in urine were measured by BQC Phenolic Quantification Assay Kit in urine of mice at different ages (6 months, 14 months, and 22 months). The values represent the mean ± SD of µg/mL. * *p* < 0.05 vs. their respective control group (evaluated using Student’s *t*-test).

**Table 1 antioxidants-12-01994-t001:** Hearing: thresholds for each frequency. Described as median and interquartile range (Q1–Q3). Mann–Whitney tests were used to compare animals in control and cocoa group at each point of time.

	Control	Cocoa	*p*-Value
**Group 6 months (dB; median (Q1–Q3))**			
4 Hz	67.5 (60–70)	60 (57.5–60)	0.004
8 Hz	50 (45–55)	45 (40–45)	0.008
12 kHz	47.5 (40–50)	40 (40–40)	0.005
16 kHz	52.5 (50–55)	40 (40–47.5)	<0.001
32 kHz	62.5 (60–65)	50 (50–55)	0.003
**Group 14 months (dB; median (Q1–Q3))**			
4 Hz	72.5 (70–77.5)	55 (50–60)	0.001
8 Hz	50 (50–50)	37.5 (30–45)	<0.001
12 kHz	50 (47.5–55)	30 (25–40)	<0.001
16 kHz	57.5 (55–60)	32.5 (30–40)	<0.001
32 kHz	70 (70–75)	50 (50–55)	<0.001
**Group 22 months (dB; median (Q1–Q3))**			
4 Hz	80 (80–80)	65 (60–65)	<0.001
8 Hz	70 (65–70)	50 (50–55)	<0.001
12 kHz	65 (60–65)	45 (45–50)	<0.001
16 kHz	70 (70–75)	55 (55–60)	<0.001
32 kHz	80 (80–85)	60 (60–70)	<0.001

**Table 2 antioxidants-12-01994-t002:** Weight. Cocoa prevents age-related loss of body weight. Animals’ body weights were recorded at 6, 14, and 22 months. Described as median and interquartile range (Q1–Q3). Mann–Whitney test were used to compare animals in control and cocoa groups at each point of time. Data are expressed as median (Q1, Q3) of body weight in grams.

Weight (Grams; Median (Q1–Q3))	Control	Cocoa	*p*-Value
6 months	31.5 (30.3–33.3)	32.0 (30.0–34.5)	0.817
14 months	31.0 (26.3–32.5)	33.3 (30.8–36.3)	0.024
22 months	27.5 (25.5–30.0)	34.5 (30.0–38.0)	<0.001

**Table 3 antioxidants-12-01994-t003:** Grip strength. Described as median and interquartile range (Q1–Q3). Mann–Whitney test was used to compare animals in control and cocoa groups at each point of time. Data are expressed as median (Q1, Q3) in grams.

Strength (Grams; Median (Q1–Q3))	Control	Cocoa	*p*-Value
6 months	99.7 (85–111.2)	91.6 (82.2–105.5)	0.581
14 months	81.5 (77.6–85.9)	90.2 (86.3–99.2)	**0.013**
22 months	71.0 (67.8–76.0)	79.0 (68.5–86.7)	0.068

**Table 4 antioxidants-12-01994-t004:** Endurance and maximum speed. Described as median and interquartile range (Q1–Q3). Mann–Whitney test was used to compare animals in control and cocoa group at each point of time. Data are expressed as median (Q1, Q3) of endurance in minutes and of maximum speed in m/min.

**Endurance** (Min; Median (Q1–Q3))	**Control**	**Cocoa**	***p*-Value**
6 months	16.4 (15.1–17.9)	14.4 (13.3–14.9)	0.059
14 months	10.0 (8.8–11.3)	17.0 (15.9–17.6)	0.003
22 months	9.0 (7.2–10.6)	13.4 (7.7–14.5)	0.004
**Maximum speed** (m/min; median (Q1–Q3))	**Control**	**Cocoa**	***p*-value**
6 months	20.0 (19.0–22.0)	18.0 (16.0–18.0)	0.061
14 months	11.5 (10.0–12.5)	17.5 (13.5–18.5)	0.010
22 months	9.0 (4.0–11.0)	16.0 (4.0–17.0)	0.001

**Table 5 antioxidants-12-01994-t005:** Frailty and % fragility components. Described as absolute and relative frequencies, *n* (%). Chi-squared test was used to compare proportions between control and cocoa group at each point in time.

	Control	Cocoa	*p*-Value
**Weight: test failure (n, %)**			
6 months	3 (25.0)	3 (25.0)	1.000
14 months	5 (41.7)	3 (25.0)	0.667
22 months	19 (82.6)	6 (26.1)	<0.001
**Strength: test failure (n, %)**			
6 months	2 (16.7)	2 (16.7)	1.000
14 months	3 (25.0)	1 (8.3)	0.590
22 months	5 (21.7)	4 (14.9)	1.000
**Endurance: test failure (n, %)**			
6 months	1 (8.0)	3 (25.0)	0.590
14 months	3 (25.0)	1 (8.3)	0.590
22 months	5 (21.7)	4 (17.4)	1.000
**Speed: test failure (n, %)**			
6 months	1 (8.0)	0 (0.0)	1.000
14 months	0 (0.0)	0 (0.0)	1.000
22 months	0 (0.0)	0 (0.0)	1.000
**Motor Coordination: test failure (n, %)**			
6 months	0 (0.0)	0 (0.0)	1.000
14 months	4 (33.3)	2 (16.7)	0.64
22 months	15 (65.2)	9 (39.1)	0.139
**Frailty Score * (%)**			
6 months	11.7	13.3	0.783
14 months	25.0	11.7	0.059
22 months	38.3	20.0	0.002

* frailty score for each age group of animals was calculated as follows: total number of test failed by the animals at each age group, divided by the total number of tests performed by these animals, expressed in percentage.

## Data Availability

Data available on request due to privacy restrictions. The data presented in this study are available on request from the corresponding author.

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
