# Peer review of "Cocoa Polyphenols Prevent Age-Related Hearing Loss and Frailty in an In Vivo Model"

_antioxidants, 2023, doi:10.3390/antiox12111994_

Round 1

Reviewer 1 Report

Comments and Suggestions for Authors

After the reading the manuscript my major concerns are as follows:

1.      Decimal system in English is based on full stops. Please replace commas with full stops in all tables and in the results section.

2.       In the legends to the Tables and Figures, please, add information on a statistical test used.

3.      Please, replace “p-valor” with “p-value” in all the Tables.

4.      Table 2 and Figure 2 – there is no need to duplicate presentations of the results in both, tabular and graphical forms. Please choose one form, either tables or figures.

5.      Figures – please, place asterisks on graphs to indicate significance.

6.      Tables – add F-statistics with degrees of freedom and the p-values, where necessary.

7.      Why the authors did not apply two-way ANOVA for analyzing repeated measure on time (treatment [control vs. cocoa] in three time points - 6, 14 and 22 months)?

8.      Figure 5. On this graph there were some outlier values. What was the procedure for eliminating such outliers? How the authors have eliminated these outliers from their set of results?

9.      Legend to figure 8. Line 388 – what exactly “SD of micro-Coulomb (ug/mL)” means?

10.   Line 429: Instead of “NW-nitro-L-arginine” it should be “N-ω-nitro-L-arginine” (Greek lowercase letter omega).

Comments on the Quality of English Language

No comments

Author Response

Thank you for the review of our manuscript entitled “Cocoa Polyphenols Prevent Age-Related Hearing Loss and Frailty in an In Vivo Model

We have reassessed it according to the recommendations of the reviewers. All changes in the text have been marked “Track Changes” where each revision occurs. We have also answered all the questions point by point to clarify the content of the manuscript.

We sincerely appreciate the comments of the reviewers and the recommendation to improve our article that we have followed.

Reviewer:

After the reading the manuscript, my major concerns are as follows:

  1. Decimal system in English is based on full stops. Please replace commas with full stops in all tables and in the results section.

Thank you for your comment. Amended, in the revised manuscript in all tables and in the results section.

  1. In the legends to the Tables and Figures, please, add information on a statistical test used.

Following the reviewer's recommendation, information has been added to the revised manuscript.

  1. Please, replace “p-valor” with “p-value” in all the Tables.

Amended, in the revised manuscript.

  1. Table 2 and Figure 2 – there is no need to duplicate presentations of the results in both, tabular and graphical forms. Please choose one form, either tables or figures.

Thanks for your observation, but we have decided not to remove Figure 2 because, as in the rest of the parameters (strength, coordination,...) the tables show the comparison between groups for each time, while the figures want to show the change experienced over time due to aging of each of the two groups, and that they are complementary.

  1. Figures – please, place asterisks on graphs to indicate significance.

Thank you for your comment. Amended, in the revised manuscript in all figures.

  1. Tables – add F-statistics with degrees of freedom and the p-values, where necessary.

In response to your request to add F-statistics with degrees of freedom in our tables, it's important to clarify that the comparisons between groups for each of the studied time points, as presented in Tables 1 to 4, were conducted using the Mann-Whitney-Wilcoxon test. The Mann-Whitney-Wilcoxon test is a non-parametric approach, similar to the Student's independent samples t-test, and its interpretation is based on differences between sample ranks. Unlike parametric tests such as T, F, or Chi-squared, this test does not involve degrees of freedom associated with a sampling distribution. However, it's worth noting that all tables include p-values for the comparisons, providing the necessary information to assess the statistical significance of the results.

  1. Why the authors did not apply two-way ANOVA for analyzing repeated measure on time (treatment [control vs. cocoa] in three time points - 6, 14 and 22 months)?

Thank you very much for your comment. While it is true that a repeated measures ANOVA is the test of choice when analyzing the effects of two or more conditions or treatments on a quantitative dependent variable measured at multiple time points, it is not suitable in our case, as different individuals have been studied at each time point. As indicated in the "Experimental Animals" section of the Materials and Methods, "After completing the tests, the 6-month-old group was sacrificed. This same procedure was performed for the 14- (24 male and female mice, 12 control group, 12 cocoa group) and 22-month-old (46 male and female mice, 23 control group, 23 cocoa group) groups." These are not repeated measurements on the same group of subjects over time. Therefore, we have chosen to conduct between-group comparisons (control vs. cocoa) at each point of time as well as and within-group comparisons for different time points, always employing tests for independent groups.

  1. Figure 5. On this graph there were some outlier values. What was the procedure for eliminating such outliers? How the authors have eliminated these outliers from their set of results?

Thank you for your question regarding the presence of outlier values in the graph. It's important to clarify that the outliers have not been removed from the analysis (the results include data from the entire set of mice studied at each time point and in each group. The outliers have only been removed from the graph itself for the purpose of visualization.

  1. Legend to figure 8. Line 388 – what exactly “SD of micro-Coulomb (ug/mL)” means?

Thanks for the observation, it was a mistake. Amended in revised manuscript.

  1. Line 429: Instead of “NW-nitro-L-arginine” it should be “N-ω-nitro-L-arginine” (Greek lowercase letter omega).

Amended, in the revised manuscript.

Reviewer 2 Report

Comments and Suggestions for Authors

In this article, the authors examine the effects of cocoa diet in mice (ages of 6, 14, and 22 months). Hearing, frailty, total antioxidant capacity and total polyphenols in urine samples were measured in mice. The authors conclude that an improvement was observed in the cocoa groups at 6, 14 and 22 months compared to the no cocoa group; furthermore, cocoa diet significantly retards the development of frailty and increases the concentration of polyphenols excreted in the urine, which increases the total antioxidant capacity. In conclusion, cocoa, due to its antioxidant properties, leads to significant protection against ARHL and frailty. 

The article is very interesting in the aging process.

Introduction: 

Line 59-63: Free radicals are essentially reactive oxygen and nitrogen species (ROS/RNS), an inevitable product of cellular respiration. When the endogenous antioxidant system is overcome either by the production of excess ROS/RNS, accumulation of toxic free radicals occurs, leading to oxidative stress-induced damage to proteins and lipids in the cytosol and cell membranes, as well as to the mitochondrial and nuclear genome”.

The authors refer to ROS/RNS, proteins and lipids damage in the cytosol and cell membranes, as well as to the mitochondrial and nuclear genome, but they do not quantify them in their study.

I suggest to expand these parts relating them to enzymatic and non-enzymatic antioxidants.

Materials and Methods 

I think, it would be interesting and useful for the reader to include an experimental diagram and if possible, some images during testing the mice: Treadmill Test, Frailty score. 

Why haven't you studied individual antioxidants after diet in mouse?

Line 227-228: pay attention to repetitions (To compare the quantitative variables at…” 

Have the correlations been analyzed? Provide analyzes if missed

Results 

To help the reader, I recommend to insert colours in the Figure (from 1 to 8).

I don't see any correlation graphs

Discussion 

Line 418-423: These findings correlate with significantly improved ASSR auditory thresholds in mice receiving treatment with cocoa at the frequencies studied (from 4 to 32 kHz), supporting a possible correlation  between the hearing data and the supposed protection of the cochlear sensory tissue. With  this study, for the first time, we demonstrate the otoprotective effect of cocoa against ARHL”.

I can't find any correlation results.

Line 446-448: “The oxidative stress, either due to lack of elimination of or overproduction of ROS, is pivotal not only for the start development of ARHL but also for triggering frailty syndrome”.

Furthermore, if ROS are so important, why haven't you measured them?

The authors also mention NO (Line 503), but this was also not measured.

It's my opinion that other preclinical studies must be necessary before considering a clinical study. Many biomarkers have not been analyzed. Many points need to be explained

Comments on the Quality of English Language

none

Author Response

Thank you for the review of our manuscript entitled “Cocoa Polyphenols Prevent Age-Related Hearing Loss and Frailty in an In Vivo Model

We have reassessed it according to the recommendations of the reviewers. All changes in the text have been marked “Track Changes” where each revision occurs. We have also answered all the questions point by point to clarify the content of the manuscript.

We sincerely appreciate the comments of the reviewers and the recommendation to improve our article that we have followed.

Reviewer:

In this article, the authors examine the effects of cocoa diet in mice (ages of 6, 14, and 22 months). Hearing, frailty, total antioxidant capacity and total polyphenols in urine samples were measured in mice. The authors conclude that an improvement was observed in the cocoa groups at 6, 14 and 22 months compared to the no cocoa group; furthermore, cocoa diet significantly retards the development of frailty and increases the concentration of polyphenols excreted in the urine, which increases the total antioxidant capacity. In conclusion, cocoa, due to its antioxidant properties, leads to significant protection against ARHL and frailty.

The article is very interesting in the aging process.

Introduction:

Line 59-63: “Free radicals are essentially reactive oxygen and nitrogen species (ROS/RNS), an inevitable product of cellular respiration. When the endogenous antioxidant system is overcome either by the production of excess ROS/RNS, accumulation of toxic free radicals occurs, leading to oxidative stress-induced damage to proteins and lipids in the cytosol and cell membranes, as well as to the mitochondrial and nuclear genome”.

The authors refer to ROS/RNS, proteins and lipids damage in the cytosol and cell membranes, as well as to the mitochondrial and nuclear genome, but they do not quantify them in their study.

I suggest to expand these parts relating them to enzymatic and non-enzymatic antioxidants.

Thanks for your comment. Amended, in the introduction section of the revised manuscript.

Materials and Methods

I think, it would be interesting and useful for the reader to include an experimental diagram and if possible, some images during testing the mice: Treadmill Test, Frailty score.

Thanks for your observation. Amended, in the revised manuscript. Also, we have included videos as supplementary material of the tests that have been done on the mice.

Why haven't you studied individual antioxidants after diet in mouse?

Thank you for your comment. The objective we set was to study the effect of a diet rich in cocoa due to its antioxidant properties due to its high polyphenol content. As explained in the introduction, cocoa is composed of different types of polyphenols and, we wanted to see the effect of the mixture of these compounds because they interact between them, and each one can act at different levels in the organism. For this reason, antioxidant compounds were not studied individually and were not determined individually in the urine, only the presence of total polyphenols was determined.

Line 227-228: pay attention to repetitions (“To compare the quantitative variables at…”

Amended, in the revised manuscript

Have the correlations been analyzed? Provide analyzes if missed

We have not conducted any correlation analysis in our study. This is because the aims of our research were to examine potential differences between the Control and Cocoa study groups over time in each of the various study variables. We oriented the statistics towards assessing these differences, and we did not intend to investigate relationships between the different variables.

However, we are open to considering the inclusion of correlation analyses if you believe it would be relevant to enhance our work. If you could specify the particular variables, you would like us to correlate, we would be happy to provide results as supplementary tables.

Results

To help the reader, I recommend to insert colours in the Figure (from 1 to 8).

Following the reviewer's recommendation, the figures have been made in color in the revised manuscript.

I don't see any correlation graphs

As previously stated, no correlation analysis between variables has been conducted.

Discussion

Line 418-423: “These findings correlate with significantly improved ASSR auditory thresholds in mice receiving treatment with cocoa at the frequencies studied (from 4 to 32 kHz), supporting a possible correlation between the hearing data and the supposed protection of the cochlear sensory tissue. With this study, for the first time, we demonstrate the otoprotective effect of cocoa against ARHL”.

I can't find any correlation results.

Thank you for your observation. The term "correlation" has been used incorrectly in the text and has been modified by relationship and association. The objective of the study was not to establish "correlations" through statistical analysis as explained previously.

Line 446-448: “The oxidative stress, either due to lack of elimination of or overproduction of ROS, is pivotal not only for the start development of ARHL but also for triggering frailty syndrome”. Furthermore, if ROS are so important, why haven't you measured them?

Thanks for your comment. In fact, we could have measured ROS in plasma, for example, but we decided better to determine the total antioxidant capacity (TAC) in plasma to evaluate the network of non-enzymatic antioxidants as a rapid and indirect measure to evaluate oxidative stress. If TAC values increase, it means that antioxidant defenses increase and, therefore, ROS levels decrease after the cocoa-rich diet.

This project has a second phase in which both antioxidant enzymes (Catalase, SOD and GPx) and ROS are analyzed in the cochlea of mice after sacrifice.

The authors also mention NO (Line 503), but this was also not measured.

In the Discussion we mention NO in the context of giving as an example a study in which the TAC measurement is used as a useful tool to determine the antioxidant capacity of different compounds and, specifically, in the study it is also measured NO levels as a marker of ROS.

As mentioned above, the next phase of the study looks at ROS levels, among other markers of stress and damage in the mouse cochlea.

It's my opinion that other preclinical studies must be necessary before considering a clinical study. Many biomarkers have not been analyzed. Many points need to be explained

Round 2

Reviewer 1 Report

Comments and Suggestions for Authors

1. Page 7, Table 1. There are still commas instead of full stops in columns (control and cocoa) presenting decimal system in English. Please, correct these values.

2. Page 8. Legend to Figure 3. The proper name of this statistical test is "Student's t-test". Student is the name of the statistician, who created this test. Replace "T-student test" with "Student's t-test".

3. Page 9. Legend to Figure 4. Please, delete the plural form of statistical tests used in this study. There is: a Mann-Whitney test, Kruskal-Wallis test, Sidak test. There are not: Sidak tests, etc. Please delete the final "s".

4. The same holds true for the Legend to Figures 5, 6, 7, 8. Please delete the final "s" in the name of the statistical test.

5. Table 4 on page 11. There are still commas instead of full stops in the COCOA column for "Endurance" presenting decimal system in English. Please, correct these values.

6. Figure 8. Too low resolution makes this figure illisible. Please, increase resolution of this figure. Legend to Figure 8. Please correct the name of the Student's t-test.

7. Figures 9 and 10 - Legends - please correct the name of the Student's t-test.

8. Table 2 on page 9. There are still commas instead of full stops in the COCOA column for "Weight" presenting decimal system in English. Please, correct these values.

Comments on the Quality of English Language

No comments

Author Response

We have reassessed it according to the recommendations of the reviewer. All changes in the text have been marked “Track Changes” where each revision occurs. We have also answered all the questions point by point to clarify the manuscript's content. We sincerely appreciate the comments of the reviewers and the recommendation to improve the article we have followed.

Reviewer:

  1. Page 7, Table 1. There are still commas instead of full stops in columns (control and cocoa) presenting decimal system in English. Please, correct these values.

Thanks for your observation. It is amended, in the revised manuscript.

  1. Page 8. Legend to Figure 3. The proper name of this statistical test is "Student's t-test". Student is the name of the statistician, who created this test. Replace "T-student test" with "Student's t-test".

Thanks for your observation. Amended, in the revised manuscript.

  1. Page 9. Legend to Figure 4. Please, delete the plural form of statistical tests used in this study. There is: a Mann-Whitney test, Kruskal-Wallis test, Sidak test. There are not: Sidak tests, etc. Please delete the final "s".

It is amended, in the revised manuscript.

  1. The same holds true for the Legend to Figures 5, 6, 7, 8. Please delete the final "s" in the name of the statistical test.

Amended, in the revised manuscript.

  1. Table 4 on page 11. There are still commas instead of full stops in the COCOA column for "Endurance" presenting decimal system in English. Please, correct these values.

Thanks for your observation. It is amended, in the revised manuscript.

  1. Figure 8. Too low resolution makes this figure illisible. Please, increase resolution of this figure. Legend to Figure 8. Please correct the name of the Student's t-test.

Amended, in the revised manuscript.

  1. Figures 9 and 10 - Legends - please correct the name of the Student's t-test.

Amended, in the revised manuscript.

  1. Table 2 on page 9. There are still commas instead of full stops in the COCOA column for "Weight" presenting decimal system in English. Please, correct these values.

Amended, in the revised manuscript.

Reviewer 2 Report

Comments and Suggestions for Authors

The manuscript improved after the revisions, however I recommend reviewing the sentences in Spanish (line 74-77).

Comments on the Quality of English Language

none

Author Response

We have reassessed it according to the recommendations of the reviewer. All changes in the text have been marked “Track Changes” where each revision occurs. We have also answered all the questions point by point to clarify the manuscript's content. We sincerely appreciate the comments of the reviewers and the recommendation to improve the article we have followed.

Reviewer:

The manuscript improved after the revisions, however I recommend reviewing the sentences in Spanish (line 74-77).

Thanks for your observation. Amended, in the revised manuscript.